

# Performance evaluation of deep neural ensembles toward malaria parasite detection in thin-blood smear images

Sivaramakrishnan Rajaraman,  Stefan Jaeger and  Sameer K. Antani

Communications Engineering Branch, National Library of Medicine, National Institutes of Health, Bethesda, MD, United States of America

## ABSTRACT

**Background**. Malaria is a life-threatening disease caused by *Plasmodium* parasites that infect the red blood cells (RBCs). Manual identification and counting of parasitized cells in microscopic thick/thin-film blood examination remains the common, but burdensome method for disease diagnosis. Its diagnostic accuracy is adversely impacted by inter/intra-observer variability, particularly in large-scale screening under resource-constrained settings.

**Introduction**. State-of-the-art computer-aided diagnostic tools based on data-driven deep learning algorithms like convolutional neural network (CNN) has become the architecture of choice for image recognition tasks. However, CNNs suffer from high variance and may overfit due to their sensitivity to training data fluctuations.

**Objective**. The primary aim of this study is to reduce model variance, improve robustness and generalization through constructing model ensembles toward detecting parasitized cells in thin-blood smear images.

**Methods**. We evaluate the performance of custom and pretrained CNNs and construct an optimal model ensemble toward the challenge of classifying parasitized and normal cells in thin-blood smear images. Cross-validation studies are performed at the patient level to ensure preventing data leakage into the validation and reduce generalization errors. The models are evaluated in terms of the following performance metrics: (a) Accuracy; (b) Area under the receiver operating characteristic (ROC) curve (AUC); (c) Mean squared error (MSE); (d) Precision; (e) F-score; and (f) Matthews Correlation Coefficient (MCC).

**Results**. It is observed that the ensemble model constructed with VGG-19 and SqueezeNet outperformed the state-of-the-art in several performance metrics toward classifying the parasitized and uninfected cells to aid in improved disease screening.

**Conclusions**. Ensemble learning reduces the model variance by optimally combining the predictions of multiple models and decreases the sensitivity to the specifics of training data and selection of training algorithms. The performance of the model ensemble simulates real-world conditions with reduced variance, overfitting and leads to improved generalization.

Corresponding author
Sivaramakrishnan Rajaraman,
sivaramakrishnan.rajaraman@nih.gov

## INTRODUCTION

Malaria is a serious and life-threatening disease caused by *Plasmodium* parasitic infection transmitted through the bite of female Anopheles mosquitoes. The parasites mature in the liver, are released into the human bloodstream and infect the red blood cells (RBCs) to result in fatal symptoms. Several species of the parasites exist including *Plasmodium falciparum, P. vivax, P. ovale, P. Knowlesi, and P. malariae*; however, *P. falciparum* can be lethal and infects the majority of global population. According to the 2018 World Health Organization (WHO) report, an estimated 435,000 malaria-related deaths are reported globally. Children under 5 years of age are reported to be the most vulnerable, accounting for 61% of the estimated death counts. The disease transmitted through *Plasmodium falciparum* has a high prevalence in Africa followed by South-East Asia and Eastern Mediterranean regions. An estimated US $3.1 billion has been reported to be invested worldwide in malaria control and elimination strategies by disease-endemic countries (*WHO, 2018*). Early diagnosis and treatment is the most effective way to prevent the disease. Microscopic thick/thin-film blood examination remains the most reliable and commonly known method for disease diagnosis (*Centers for Disease Control and Prevention, 2018*). However, manual diagnosis is a burdensome process, the diagnostic accuracy is severely impacted by the liability imposed by factors including inter/intra-observer variability and large-scale screening, particularly in disease-endemic countries with resource-constrained settings (*Mitiku, Mengistu & Gelaw, 2003*).

Computer-aided diagnostic (CADx) tools using machine learning (ML) algorithms applied to microscopic blood smear images have the potential to reduce clinical burden by assisting with triage and disease interpretation. *Poostchi et al. (2018)* provided a survey on such techniques. These tools process medical images for typical appearances and highlight pathological features to supplement clinical decision-making. For these reasons, CADx tools have gained prominence in image-based medical diagnosis and risk assessment. However, a majority of these tools applied to malaria diagnosis use handcrafted feature extraction algorithms that are optimized for individual datasets and trained for specific variability in source machinery, dimension, position, and orientation of the region of interest (ROI) (*Ross et al., 2006*; *Das et al., 2013*). At present, data-driven deep learning (DL) methods have superseded the performance of handcrafted feature extraction mechanisms by self-discovering the attributes from raw pixel data and performing end-to-end feature extraction and classification (*LeCun, Bengio & Hinton, 2015*). In particular, convolutional neural networks (CNN), a class of DL models, have demonstrated promising results in image classification, recognition, and localization tasks (*Krizhevsky, Sutskever & Hinton, 2012*; *Redmon et al., 2016*).

The promising performance of CNNs is attributed to the availability of huge amounts of annotated data. Under circumstances of limited data availability as in the case of medical images, transfer learning strategies are adopted. The CNN models are pretrained on large-scale datasets like ImageNet (*Deng et al., 2009*) to transfer the knowledge learned in the form of generic image features to be applied for the target task. The pretrained weights
serve as a good initialization and are found to perform better than training the model from scratch with randomly initialized weights.

Literature studies reveal the application of conventional ML and data-driven DL methods toward the challenge of malaria parasite detection in thin-blood smear images. *Dong et al. (2017)* compared the performance of kernel-based algorithms like support vector machine (SVM), and CNNs toward classifying infected and normal cells. A small-scale collection of segmented RBCs were randomly split into train/validation/test sets. It was observed that the CNNs achieved a classification accuracy of over 95% and significantly outperformed the SVM classifier that obtained 92% accuracy. The CNNs self-discovered the features from the raw pixel data, thereby requiring minimal human intervention for automated diagnosis. *Liang et al. (2017)* performed cross-validation studies at the cell level to evaluate the performance of custom and pretrained CNN models toward classifying parasitized and normal cell images. Experimental results demonstrated that the custom CNN outperformed the pretrained AlexNet (*Krizhevsky, Sutskever & Hinton, 2012*) model with an accuracy of 97.37%. In another study (*Bibin, Nair & Punitha, 2017*), the authors performed randomized splits with peripheral smear images and evaluated the performance of a shallow deep belief network toward detecting the parasites. Experimental results demonstrated that the deep belief network showed promising performance with an F-score of 89.66% as compared to that of SVM based classification that gave an F-score of 78.44%. *Gopakumar et al. (2018)* developed a customized CNN to analyze a focal stack of slide images for the presence of parasites. In the process, they observed that the custom CNN model achieved a Matthews Correlation Coefficient (MCC) score of 98.77% and considerably outperformed the SVM classifier that achieved 91.81% MCC. These studies were evaluated at the cell level, with randomized splits and/or small-scale datasets. The reported outcomes are promising; however, patient-level cross-validation studies with a large-scale clinical dataset are required to substantiate their robustness and generalization to real-world applications. *Rajaraman et al. (2018a)* used a large-scale, annotated clinical image dataset, extracted the features from the optimal layers of pretrained CNNs and statistically validated their performance at both cell and patient level toward discriminating parasitized and uninfected cells. It was observed that at the patient level, the pretrained ResNet-50 model outperformed the other CNNs with an accuracy of 95.9%. However, deep neural networks learn through stochastic optimization and are limited in performance due to their high variance in predictions that arises due to their sensitivity to small fluctuations in the training set. This results in modeling the random noise from the training data and leads to overfitting. This variance is frustrating especially during model deployment. An effective approach to reducing the variance is to train multiple, diverse models and combine their predictions. The process results in ensemble learning that leads to predictions that are better than any individual model (*Dietterich, 2000*). DL models and ensemble learning are known to deliver inherent benefits of non-linear decision making, the combination of these strategies could effectively minimize variance and enhance learning.

Ensemble learning strategies are often applied to obtain stable and promising model predictions. *Krizhevsky, Sutskever & Hinton (2012)* used a model averaging ensemble to achieve state-of-the-art performance in the ImageNet Large Scale Visual Recognition

Competition (ILSVRC) 2012 classification task. Model ensembles are used by the winning teams in Kaggle and other machine learning challenges. Literature studies show that ensemble methods have been applied to medical image classification tasks. In (*Lakhani & Sundaram, 2017*), the authors evaluated the efficacy of an ensemble of custom and pre-trained deep CNNs toward Tuberculosis (TB) detection in chest X-rays (CXRs). It was observed that the ensemble accurately detected the disease with an AUC of 0.99. *Rajaraman et al. (2018b)* created a stacking of classifiers operating with handcrafted and deep CNN features toward improving TB detection in CXRs. The performance of the individual models and the model ensemble was evaluated on four different CXR datasets. It was observed that the model ensemble outperformed the individual models and the state-of-the-art, with an AUC of 0.991, 0.965, 0.962, and 0.826 with Shenzhen, Montgomery, India, and Kenya CXR datasets respectively. However, to our knowledge, there is no available literature on the application and evaluation of ensemble methods toward malaria parasite detection in thin-blood smear images.

In this study, we evaluate the performance of custom and pretrained CNNs and construct an optimal ensemble model to deliver predictions with reduced bias and improved generalization toward the challenge of classifying parasitized and normal RBCs in thin-blood smear images. Compared to our previous study (*Rajaraman et al., 2018a*), we aim to reduce the model variance by combining the predictions of multiple models and reduce model sensitivity to the specifics of training instances and selection of training methods. In the process, the model ensemble is expected to demonstrate improved performance and generalization, and simulate real-world conditions with reduced variance. To the best of our knowledge, this is the first study to construct and statistically evaluate an ensemble model to classify a large-scale clinical dataset of parasitized and uninfected cells toward the current task.

## MATERIALS & METHODS

### Data Collection and preprocessing

The parasitized and normal cell image collection used in this study is made publicly available by *Rajaraman et al. (2018a)*. Giemsa-stained thin-blood smear slides were collected from *P. falciparum*-infected patients and healthy controls and photographed using a smartphone camera. The slide images were manually annotated by an expert, de-identified, and archived. The Institutional Review Board (IRB) at the National Library of Medicine (NLM), National Institutes of Health (NIH) granted approval to carry out the study within its facilities (IRB# 12972). A level-set based algorithm was applied to detect and segment the RBCs (*Rajaraman et al., 2018a*). The dataset includes 27,558 cell images with equal instances of parasitized and healthy RBCs. Cells containing *Plasmodium* are labeled as positive samples while normal instances contain no *Plasmodium* but other objects including impurities and staining artifacts. The images are re-sampled to $100 \times 100$ pixel dimensions and mean normalized for faster model convergence.

## Predictive models and computational resources

We evaluated the performance of the following CNN models toward the task of classifying parasitized and uninfected RBCs segmented from thin-blood smear images: (a) custom CNN; (b) VGG-19 (*Simonyan & Zisserman, 2015*); (c) SqueezeNet (*Iandola et al., 2016*); and (d) InceptionResNet-V2 (*Szegedy, Ioffe & Vanhoucke, 2016*). The VGG-19 model is developed by the Oxford's Visual Geometry Group. It uses $3\times3$ filters throughout its depth and offers 7.3% test classification error in the ILSVRC-2012 classification challenge. The model is also known to generalize well to other datasets. The SqueezeNet model offers comparable accuracy of AlexNet in the ILSVRC challenge at a reduced computational cost. The model uses $1\times1$ filters and a squeezing layer to reduce the depth and parameters to offer high-level feature abstraction. The InceptionResNet-V2 model combines the architectural benefits of Inception modules and residual blocks to achieve top accuracy in the ILSVRC image classification benchmark. The predictive models are evaluated through five-fold cross-validation. We performed cross-validation at the patient level and ensured preventing data leakage into the validation to reduce generalization errors. The images are augmented with transformations to prevent overfitting of the training data and improve model generalization and performance. Augmentations including rotations, translation, shearing, zooming, and flipping are performed on-the-fly during model training. The models are evaluated in terms of the following performance metrics: (a) Accuracy; (b) Area under the receiver operating characteristic (ROC) curve (AUC); (c) Mean squared error (MSE); (d) Precision; (e) F-score; and (f) Matthews Correlation Coefficient (MCC).

We trained the models on an Ubuntu system with Xeon E5-2640v3 processor, 64GB Random Access Memory (RAM), and NVIDIA® 1080Ti graphical processing unit (GPU). The models are configured in Python using Keras API with Tensorflow backend and CUDA/CuDNN dependencies for GPU acceleration.

## Configuring custom and pretrained CNN models
### Custom model configuration

The custom CNN model has three blocks of batch normalization, convolution, pooling, and convolutional dropout layers. The convolutional layers use $5\times5$ filters throughout the depth and same values for padding to maintain identical feature map dimensions. The first convolutional layer has 64 filters, the number of filters is doubled after every max-pooling layer. Usage of batch normalization layers reduces overfitting and improves generalization by normalizing the output of the previous activation layers. Non-linear activation layers add non-linearity into the decision-making process and speed up training and convergence (*Shang et al., 2016*). Usage of $2\times2$ max-pooling layers summarizes the outputs of the neural groups in the feature maps from the previous layers. The dropout used in the convolutional layers offers regularization by constraining the model adaptation to the training data and avoiding overfitting (*Srivastava et al., 2014*). The output of the deepest convolutional layer following a dropout is fed to a global average pooling (GAP) layer that performs dimensionality reduction by producing the spatial average of the feature maps to be fed into the first dense layer. The output of the dense layer following a dropout

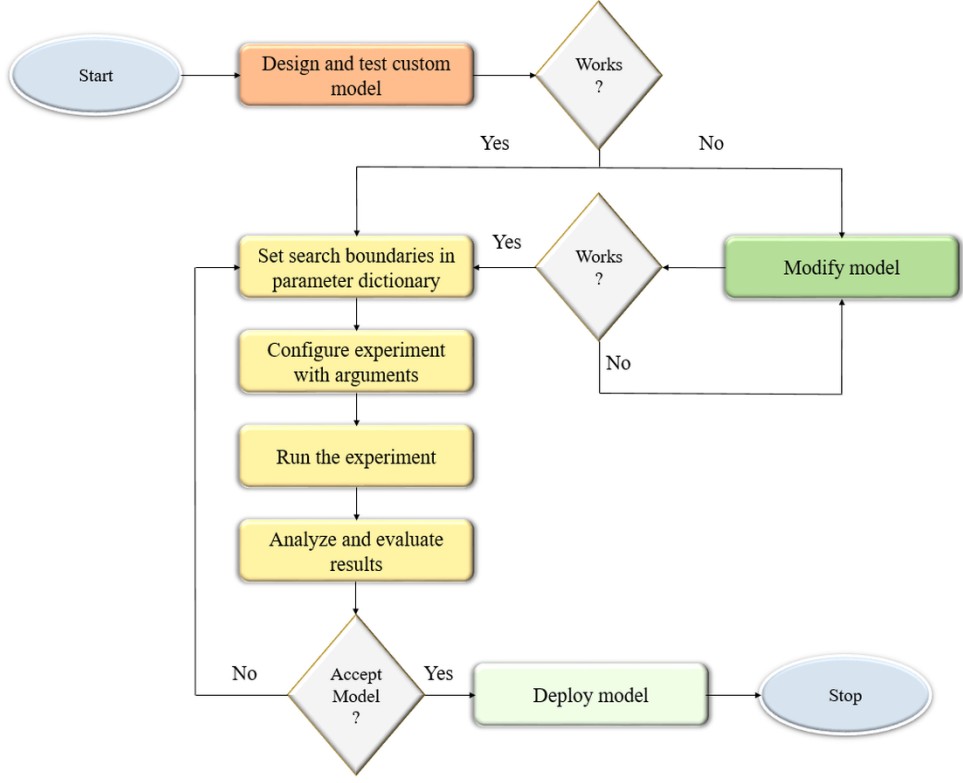

**Figure 1 Process flow diagram for optimizing the hyperparameters of the custom CNN model.** The model architecture is instantiated and evaluated within the parameter search boundaries. The process is repeated until an acceptable model is found.

is fed into the final dense layer with two neurons. The model is trained and optimized to minimize the cross-entropic loss and output the probability of predictions.

We optimized the parameters and hyperparameters of the custom CNN model using the Talos optimization tool (https://github.com/autonomio/talos). The process flow diagram of the optimization procedure is shown in Fig. 1. Talos incorporates random, grid, and probabilistic hyperparameter optimization strategies that helps to maximize the model efficiency and performance on the target tasks. The model architecture is instantiated and evaluated within the search boundaries set in the parameter dictionary. The following parameters are optimized: (a) dropout in the convolutional layer; (b) dropout in the dense layer; (c) optimizer; (d) activation function; and (e) number of neurons in the dense layer. The search ranges for the optimizable parameters are shown in Table 1. The process is repeated until an acceptable model is found.

## Fine-tuning the pretrained CNN models

We instantiated the pretrained CNNs with their convolutional layer weights and truncated these models at their deepest convolutional layer. A GAP and dense layer are added to learn from and predict on the cell image data. The generalized block diagram of the usage of pretrained models is shown in Fig. 2.
**Table 1  Search ranges for the hyperparameters of the custom CNN model.** The following parameters are optimized: (A) Dropout in the convolutional layer; (B) Dropout in the dense layer; (C) Optimizer; (D) Activation function; and (E) Number of neurons in the dense layer.

| Parameters | Search range |
|---|---|
| Convolutional dropout | [0.25, 0.5] |
| Dense dropout | [0.25, 0.5] |
| Optimizer | [SGD, Adam] |
| Activation | [ReLU, eLU] |
| Dense neurons | [256, 512] |

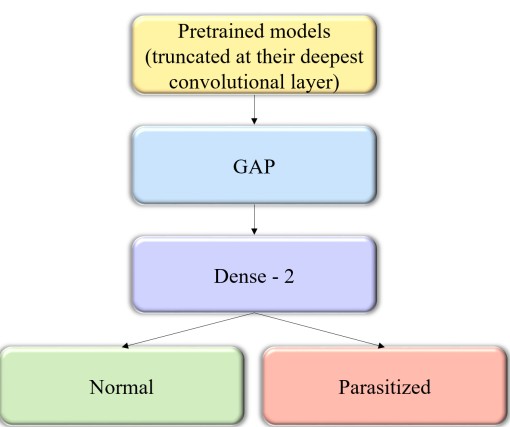

**Figure 2  The custom architecture of pretrained models used in this study.** The pretrained CNNs are instantiated with their convolutional layer weights, truncated at their deepest convolutional layer, and added with a GAP and dense layer.

We fine-tuned the models entirely using a very low learning rate (0.0001) with the Adam optimizer to minimize the categorical cross-entropic loss as not to rapidly modify the pretrained weights.

## Constructing the model ensemble

The predictions of the custom and pretrained CNN models are averaged to construct the model ensemble. Figure 3 shows the process flow diagram for combining the predictions of the predictive models and selecting the optimal ensemble from a collection of model combinations, for further deployment. The inherent benefit of the model averaging ensemble is that it needs no training since averaging the predictions does not take any learnable parameters.

## Statistical analysis

We performed statistical analyses to ensure that the results are correctly interpreted and apparent relationships are significant. Statistical tests assist in evaluating the statistically significant difference in the performance of the individual models, called the base-learners, and model ensembles. We empirically determined the presence/absence of a statistically significant difference in the mean values of the performance metrics of the pretrained

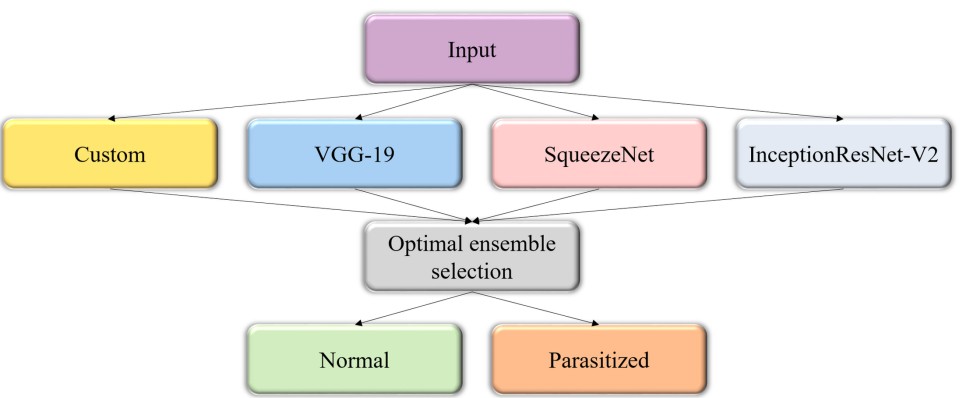

**Figure 3** **Process flow diagram depicting the construction of the model averaging ensemble.** The averaging ensemble averages the prediction probabilities from the individual models.

and ensemble models. We opted to perform a one-way analysis of variance (ANOVA) to determine the existence of a statistically significant difference (*Rossi, 1987*). However, one-way ANOVA should be performed only when the following assumptions are satisfied: (a) data normality; (b) homogeneity of variances; (c) absence of significant outliers; and (d) independence of observations (*Daya, 2003*). We performed the Shapiro–Wilk test (*Shapiro & Wilk, 1965*) to investigate for the normal distribution of data and Levene's test (*Gastwirth, Gel & Miao, 2009*), for the homogeneity of variances. We also analyzed the data for the presence of significant outliers. The null hypothesis is accepted if there exists no statistically significant difference in the performance metrics. If the test returns a statistically significant result ($p < 0.05$), the null hypothesis is rejected and the alternate hypothesis is accepted. This demonstrates the fact a statistically significant difference in the mean values of the performance metrics exists between at least two models under study. One-way ANOVA is an omnibus test that further needs post-hoc analyses to identify the specific models that demonstrate statistically significant differences (*Kucuk et al., 2016*). We performed Tukey post-hoc analysis to identify the specific models that demonstrate these statistically significant difference in their performances. We used the IBM SPSS (IBM, Armonk, NY, USA) package to perform these analyses.

## RESULTS

### Custom model hyperparameter optimization
The performance of the optimized custom CNN and pretrained models are evaluated toward the challenge of classifying parasitized and uninfected cells. The optimal values for the parameters and hyperparameters of the custom CNN, obtained with the Talos optimization tool are shown in Table 2. The model is trained and optimized to minimize the cross-entropic loss and categorize the cell images to their respective classes.
**Table 2 Optimal hyperparameter values obtained with Talos optimization for the custom CNN model.** The custom model is trained and optimized with the hyperparameter values obtained through Talos optimization to categorize the cell images to their respective classes.

| Parameters | Optimal values |
| --- | --- |
| Convolutional dropout | 0.25 |
| Dense dropout | 0.5 |
| Optimizer | Adam |
| Activation | ReLU |
| Dense neurons | 256 |

## Evaluation of performance metrics

Table 3 lists the values for the performance metrics obtained with the custom, pretrained and the All-Ensemble models, in terms of mean and standard deviation across the cross-validated folds. The learning rate is reduced whenever the validation accuracy ceased to improve. The training and validation losses decreased with epochs that stood indicative of the learning process. It is observed that the VGG-19 model outperformed the other models in all performance metrics. The generic image features learned from the ImageNet data served as a good initialization as compared to random weights in the custom CNN model. The model is trained end-to-end to learn parasitized and normal cell-specific features to reduce bias and improve generalization. The architectural depth of the VGG-19 appeared optimal for the current task. The other pretrained models are progressively more complex and did not perform as well as the VGG-19 model. The All-Ensemble model shared the same input as the individual models and computed the average of the models' predictions. We observed that the All-Ensemble model didn't outperform VGG-19. An ensemble model yields promising predictions only when there is significant diversity among the individual base-learners (*Opitz & Maclin, 1999*). Under current circumstances, the All-Ensemble model didn't deliver promising results compared to custom and pretrained models under study. For this reason, we created several combinations of models as listed in Table 4 and averaged their predictions toward creating the optimal model ensemble for the current task. Table 5 lists the performance metrics of these model combinations. It is observed that the Ensemble-D model created with VGG-19 and SqueezeNet, outperformed the individual models and other ensembles in all performance metrics. This model combination has a significant diversity that resulted in reduced correlation in their predictions and variance to offer improved performance and generalization.

We performed the Shapiro–Wilk test to investigate for data normality and Levene's test, for homogeneity of variances. We observed that $p > 0.05$ (95% confidence interval (CI) for both tests that signified that the assumptions of data normality and homogeneity of variances are not violated. The independence of observation existed and no significant outliers are observed. Hence, we performed one-way ANOVA to explore the presence/absence of a statistically significant difference in the mean values of the performance metrics for the models. Table 6 shows the consolidated results of Shapiro–Wilk, Levene, one-way ANOVA, and Tukey post-hoc analyses. We used M1, M2, and M3, to denote the VGG-19, All-Ensemble, and Ensemble-D models respectively.

**Table 3** **Performance metrics of individual models and model ensemble.** The performance of the models are evaluated with metrics including accuracy, AUC, MSE, precision, F-score, and MCC.

| Model | Accuracy | AUC | MSE | Precision | F-score | MCC |
|---|---|---|---|---|---|---|
| Custom CNN | 99.09 ± 0.08 | 99.3 ± 0.6 | 00.9 ± 0.1 | 99.56 ± 0.1 | 99.08 ± 0.1 | 98.18 ± 0.1 |
| VGG-19 | **99.32 ± 0.1** | **99.31 ± 0.7** | **00.69 ± 0.1** | **99.71 ± 0.2** | **99.31 ± 0.1** | **98.62 ± 0.2** |
| SqueezeNet | 98.66 ± 0.1 | 98.85 ± 0.3 | 1.41 ± 0.2 | 99.44 ± 0.1 | 98.64 ± 0.1 | 97.32 ± 0.1 |
| InceptionResNet-V2 | 98.79 ± 0.1 | 99.2 ± 0.9 | 1.88 ± 0.9 | 99.56 ± 0.2 | 98.77 ± 0.1 | 97.59 ± 0.2 |
| All-Ensemble | 99.11 ± 0.1 | 98.94 ± 0.3 | 0.78 ± 0.1 | 99.67 ± 0.1 | 99.1 ± 0.1 | 98.21 ± 0.2 |

Notes.
Bold text indicates the performance measures of the best-performing model/s.

**Table 4** **Combining different models to determine the optimum ensemble.** Several combinations of models are created and their prediction probabilities are averaged in an attempt to find the best performing ensemble toward the current task.

| Combination index | Models |
|---|---|
| A | [Custom CNN, VGG-19] |
| B | [Custom CNN, SqueezeNet] |
| C | [Custom CNN, InceptionResNet-V2] |
| D | [VGG-19, SqueezeNet] |
| E | [VGG-19,InceptionResNet-V2] |
| F | [SqueezeNet, InceptionResNet-V2] |
| G | [Custom CNN,VGG-19, SqueezeNet] |
| H | [Custom CNN, VGG-19, InceptionResNet-V2] |
| I | [VGG-19, SqueezeNet,InceptionResNet-V2] |

**Table 5** **Performance metrics achieved with different combinations of model ensembles.** The performance of the different combination of model ensembles is evaluated with metrics including accuracy, AUC, MSE, precision, F-score, and MCC.

| Combination index | Accuracy | AUC | MSE | Precision | F-score | MCC |
|---|---|---|---|---|---|---|
| A | 99.34 ± 0.1 | 99.07 ± 0.5 | 0.71 ± 0.1 | 99.76 ± 0.1 | 99.32 ± 0.1 | 98.65 ± 0.2 |
| B | 98.98 ± 0.1 | 99.76 ± 0.1 | 1.07 ± 0.1 | 99.43 ± 0.1 | 98.96 ± 0.1 | 97.95 ± 0.2 |
| C | 98.72 ± 0.8 | 98.64 ± 1.1 | 1.88 ± 0.6 | 99.56 ± 0.1 | 99.07 ± 0.1 | 98.15 ± 0.2 |
| D | **99.51 ± 0.1** | **99.92 ± 0.1** | **0.63 ± 0.1** | **99.84 ± 0.1** | **99.5 ± 0.1** | **99.0 ± 0.2** |
| E | 99.16 ± 0.1 | 99.18 ± 0.2 | 0.83 ± 0.1 | 99.73 ± 0.1 | 99.15 ± 0.1 | 98.31 ± 0.2 |
| F | 98.73 ± 0.1 | 99.2 ± 0.6 | 1.65 ± 0.4 | 99.63 ± 0.2 | 99.08 ± 0.1 | 98.18 ± 0.2 |
| G | 99.21 ± 0.1 | 98.98 ± 0.2 | 0.81 ± 0.1 | 99.64 ± 0.1 | 99.2 ± 0.1 | 98.42 ± 0.1 |
| H | 99.22 ± 0.1 | 99.89 ± 0.1 | 0.82 ± 0.1 | 99.75 ± 0.1 | 99.21 ± 0.1 | 98.44 ± 0.2 |
| I | 99.13 ± 0.1 | 99.67 ± 0.1 | 0.99 ± 0.1 | 99.75 ± 0.1 | 99.12 ± 0.1 | 98.26 ± 0.2 |

Notes.
Bold text indicates the performance measures of the best-performing model/s.

Tukey post-hoc analysis is performed to determine the specific models demonstrating the statistically significant difference in performance. It is observed that a statistically significant difference in the mean values of the performance metrics existed between these models. For accuracy, the post-hoc analysis revealed that the accuracy of VGG-19 ($0.993 \pm 0.0008$, $p < 0.05$) and the All-Ensemble model ($0.991 \pm 0.0008$, $p < 0.05$) is

**Table 6** **Consolidated results of Shapiro–Wilk, Levene, one-way ANOVA and Tukey post-hoc analyses.** The value $p > 0.05$ (95% CI) for Shapiro-Wilk and Levene's test signified that the assumptions of data normality and homogeneity of variances are not violated. Hence, one-way ANOVA analysis is performed to explore the presence/absence of a statistically significant difference in the mean values of the performance metrics for the models.

| Metric | Shapiro–Wilk test ($p$) | Levene's test ($p$) | ANOVA | | Tukey post-hoc ($p < 0.05$) |
|---|---|---|---|---|---|
| | | | F | $p$ | |
| Accuracy | 0.342 | 0.316 | 37.151 | $p < 0.05$ | (M1, M2, M3) |
| AUC | 0.416 | 0.438 | 8.321 | $p < 0.05$ | (M2, M3) |
| MSE | 0.862 | 0.851 | 11.288 | $p < 0.05$ | (M1, M2) & (M2, M3) |
| Precision | 0.52 | 0.294 | 5.841 | $p < 0.05$ | (M2, M3) |
| F-score | 0.599 | 0.73 | 34.799 | $p < 0.05$ | (M1, M2) & (M1, M3) |
| MCC | 0.63 | 0.697 | 35.062 | $p < 0.05$ | (M1, M2, M3) |

statistically significantly lower compared to the Ensemble-D model ($0.995 \pm 0.0005$). A similar trend is observed for AUC where the AUC for Ensemble-D model and VGG-19 is $0.9993 \pm 0.0004$) and $0.993 \pm 0.0006$, $p > 0.05$) respectively. Considering the harmonic mean of precision and recall as demonstrated by the F-score, the Ensemble-D model ($0.995 \pm 0.001$) outperformed the All-Ensemble ($0.991 \pm 0.0009$, $p < 0.05$) and VGG-19 ($0.993 \pm 0.0008$, $p < 0.05$) models. Similar trends are observed for the performance metrics including MCC, precision, and MSE. The Ensemble-D model statistically significantly outperformed the VGG-19 and All-Ensemble model in all performance metrics.

**Web-based model deployment**

We deployed the trained model into a web browser to enable running the model at reduced computational cost and alleviate issues due to the complex backend, architecture pipelines, and communication protocols. A snapshot of the web application is shown in Fig. 4. The benefits of running the model on web browsers include (a) privacy, (b) low-latency, and (c) cross-platform implementation (*Manske & Kwiatkowski, 2009*). Client-side models facilitate privacy while dealing with sensitive data, not supposed to be transferred to the server for inference. Low latency is achieved by reducing the client–server communication overhead. Client-side networks offer cross-platform benefits by working on the web browser irrespective of the type of the operating system. It does not demand installation of libraries and drivers to perform inference. We used TensorflowJS to convert the model to layer API format. Express for NodeJS is used to set up the web server, serve the model and host the web application. Express offers the web framework and NodeJS is the open-source run-time environment that executes JavaScript code on the server-side. The node program starts the server and hosts the model and supporting files. We named the application as *Malaria Cell Analyzer App* to which the user submits an image of the parasitized/uninfected cell and the model embedded into the browser gives the predictions.

## DISCUSSIONS

We ensured that the custom CNN converges to an optimal solution through (a) architecture and hyper-parameter optimization, (b) implicit regularization imposed by batch normalization, and (c) improved generalization through aggressive dropouts in

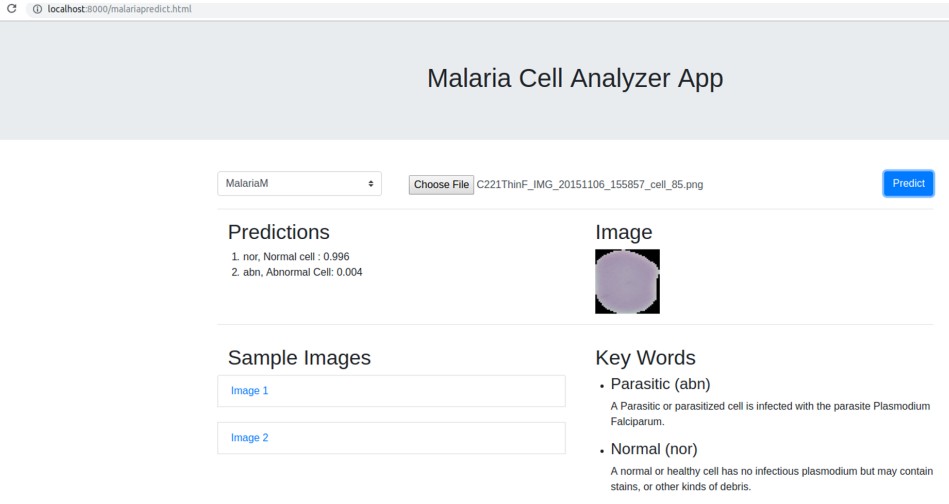

**Figure 4** **Snapshot of the web application interface.** The web application is placed into the static web directory and the web server is initiated to browse through the *Malaria Cell Analyzer App*. The user submits a cell image and the model embedded into the browser gives the predictions.

the convolutional and dense layers. We performed cross-validation at the patient-level to present a realistic performance measure of the predictive models so that the test data represents truly unseen images with no leakage of information pertaining to the staining variations or other artifacts from the training data.

CNNs suffer from the limitation of high variance as they are highly dependent on the specifics of the training data and prone to overfitting leading to increased bias and reduced generalization. We addressed this issue by training multiple models to obtain a diverse set of predictions that can be combined to deliver optimal solutions. However, it is imperative to select diversified base-learners that are accurate in diverse regions in the feature space. For this reason, we evaluated a selection of model combinations and empirically determined the best model combination to construct the ensemble for the current task. Experimental results are statistically significant for a given statistical significance level if they are not attributed to chance and a relationship actually exists. We performed statistical analyses to determine the existence of a statistically significant difference in the performance metrics of the individual and ensemble models under study. We also performed post-hoc analyses to identify the specific models demonstrating these statistically significant performance differences.

Table 7 gives a comparison of the results achieved in this study with the state-of-the-art.

It is observed that the ensemble model constructed with VGG-19 and SqueezeNet outperformed the other models and the state-of-the-art toward classifying the parasitized and uninfected cells to aid in improved disease screening.

## CONCLUSIONS

It is observed that model ensemble using multiple DL models obtained promising predictive performance that could not be accomplished by any of the individual, constituent models.

**Table 7 Comparison of the results obtained with the proposed ensemble and the state-of-the-art literature.** The ensemble model constructed with VGG-19 and SqueezeNet outperformed the other models and the state-of-the-art toward classifying the parasitized and uninfected cells to aid in improved disease screening.

| Method | Accuracy | AUC | MSE | Precision | F-score | MCC |
|---|---|---|---|---|---|---|
| Proposed Ensemble (patient level) | **99.5** | **99.9** | **0.63** | **99.8** | **99.5** | **99.0** |
| *Rajaraman et al. (2018a)* (patient level) | 95.9 | 99.1 | – | – | 95.9 | 91.7 |
| *Gopakumar et al. (2018)* | 97.7 | – | – | – | – | 73.1 |
| *Bibin, Nair & Punitha (2017)* | 96.3 | – | – | – | – | – |
| *Dong et al. (2017)* | 98.1 | – | – | – | – | – |
| *Liang et al. (2017)* | 97.3 | – | – | – | – | – |
| *Das et al. (2013)* | 84.0 | – | – | – | – | – |
| *Ross et al. (2006)* | 73.0 | – | – | – | – | – |

**Notes.**
Bold text indicates the performance measures of the best-performing model/s.

Ensemble learning reduces the model variance by optimally combining the predictions of multiple models and decreases the sensitivity to the specifics of training data and training algorithms. We also developed a web application by deploying the ensemble model into a web browser to avoid the issues of privacy, low-latency, and provide cross-platform benefits. The performance of the model ensemble simulates real-world conditions with reduced variance, overfitting and leads to improved generalization. We believe that the results proposed are beneficial toward developing clinically valuable solutions to detect and differentiate parasitized and uninfected cells in thin-blood smear images.

### Funding

This work was supported by the Intramural Research Program of the National Library of Medicine (NLM), National Institutes of Health (NIH) and the Lister Hill National Center for Biomedical Communications (LHNCBC). The intramural research scientists (authors) at the NIH dictated study design, data collection, data analysis, decision to publish and preparation of the manuscript.

### Grant Disclosures

The following grant information was disclosed by the authors:
Intramural Research Program of the National Library of Medicine (NLM).
National Institutes of Health (NIH).
The Lister Hill National Center for Biomedical Communications (LHNCBC).

### Competing Interests

The authors declare there are no competing interests.

## Author Contributions

- Sivaramakrishnan Rajaraman conceived and designed the experiments, performed the experiments, analyzed the data, contributed reagents/materials/analysis tools, prepared figures and/or tables, authored or reviewed drafts of the paper, approved the final draft.
- Stefan Jaeger contributed reagents/materials/analysis tools, authored or reviewed drafts of the paper, approved the final draft.
- Sameer K. Antani conceived and designed the experiments, contributed reagents/-materials/analysis tools, authored or reviewed drafts of the paper, approved the final draft.

## Ethics

The following information was supplied relating to ethical approvals (i.e., approving body and any reference numbers):

The Institutional Review Board (IRB) at the National Library of Medicine (NLM), National Institutes of Health (NIH) granted approval to carry out the study within its facilities (IRB#12972).

## Data Availability

Codes are available at

https://github.com/sivaramakrishnan-rajaraman/Deep-Neural-Ensembles-toward-Malaria-Parasite-Detection-in-Thin-Blood-Smear-Images.

Data is available at https://ceb.nlm.nih.gov/repositories/malaria-datasets/.

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
