# Peer review of "Performance evaluation of deep neural ensembles toward malaria parasite detection in thin-blood smear images"

_PeerJ, doi:10.7717/peerj.6977_

## Round 0.1 · original submission · Major Revisions

I agree with the reviewers that this is an interesting paper. I also agree that it needs some changes. Please properly address the comments of the reviewers and my comments:
1. The introduction is too long. We don't need an incomplete review of deep learning. Be to the point and provide relevant info for this paper.
2. The 'major contributions' are all invalid. Please remove. These are not research questions or hypothesis. A quick read tells me that you do again DL on the same data as before in your PeerJ paper of last year. Clearly tell that in your introduction and make clear why the method in that paper requires improvement. The achieved performance increase is rather little and on an AUC basis insignificant. The question is: what do you claim to improve the performance compared to your previous paper? That would be the currently missing claim/hypothesis. So, I expect a formulation at the last paragraph like ''ensemble learning improves performance''. Then you do an experiment with and without ensemble learning and show that it improves results. Simple as that.
3. Please remove the ''the paper is organized as follows statement'' this is obvious.
4. Tell a bit more about the data. At the bare minimum, I'd expect data size and a bit about the collection process.
5. Fig 5 -11 are way too much information and I question the relevance. The tSNE plots look pretty, but I fail to see how they support your conclusions. Suggest removing them.
6. In the conclusions start with a statement on whether your hypothesis (ensemble) is helpful. Remove the conclusion that it outperforms the state-of-the-art because that is not shown yet.
7. Please re-read the ICMJE guidelines, as many of my comments above are clearly addressed and explained.

Reviewer 1 ·

Basic reporting

Clear and unambiguous, professional English used throughout.
Literature references, sufficient field background/context provided although more recent literature (2018) should be included
Figures and tables are appropriate and complement the text. More self-descriptive caption should be included.
Self-contained with relevant results to hypotheses

Experimental design

Original primary research within Aims and Scope of the journal.
The article clearly states the contributions of the article however the research question is not so clear, is this a review of DL ,methodologies or are you presenting a new method?

Rigorous investigation performed to a high technical & ethical standard including a deep analysis of the outcomes.

Methods described with sufficient detail & information to replicate.

Validity of the findings

Novel comparison of methodologies and novel framework with lot of details for a replication. Also a web-based model is proposed.
Data is available on-line from a previous work and the statistical analysis is well done.
Conclusions and contributions are well stated

Additional comments

The paper is really interesting and a great analysis of the state of the art on the use of DL for malaria detection. However I missed more recent reference since most of them are from early's 2000.
The captions from the figures and tables are in general poor, it would be easier to follow if they were self-explanatory.
Line 85 proposes protocols from WHO 2010, why not WHO 2018?
Can you give some references for this sentence: "2015). In particular, convolutional neural networks (CNN), a class of DL models have demonstrated promising results in image classification, recognition, and localization tasks"?
Between lines 125 and 140 you compare CNN with other methods and give the accuracy of the CNN approach, what is the performance of the other methods such as SVM? It is not clear the improvement achieved.
Lines 142-143 presents your previous article, could you make clear what is the novelty in this work?
In the Material and Methods section I am missing a more general overview of the whole process since sometimes is difficult to follow.. A flowchart would help. Indeed I am not sure if figure 2 is presenting this but the steps in figure 2 do not correspond with subsections. In fact Figure 2 is not well explained in the text
Line 232 could be a break line.
Line 283 states that you used both custom and pretrained models, this is not clear until this moment. Giving and explanation of the whole process at the beginning of the section will also clarify this.
At the beginning of the results section you could also states what you want to analyse
Tables 4 and 5 can be fuse in one table.
Line 435, what are you losing?
Rewrite sentence in line 436.
Figure 11 and table 6 can be fuse.
I don't understand what you are trying to show in table 7
Which model are you using in the web-based model?
Where can we find the web-based model?
Table 8 should be include in the results but not in the conclusions.
Clarify sentence in line516

·

Basic reporting

Please check overall summary (below).

Experimental design

Please check overall summary (below).

Validity of the findings

Please check overall summary (below).

Additional comments

The authors have evaluated the performance of model ensembles in an attempt to reduce variance and optimally combine the models’ predictions toward classifying parasitized and uninflected cells in thin-blood smear images. The study demonstrates significant improvement in performance, compared to the state of the art and would appeal to the research community. However the authors have to clarify on a few aspects pertaining to the methods, results, and related discussions.

Specific comments:
1. How did the authors detect and segment the red blood cells? How did they evaluate the segmentation accuracy?
2. How did the performance of the proposed ensemble compare to that of the human readers?
3. How effectively could the proposed ensemble be deployed into mobile devices or cloud for the ease of predictions?
4. How did the authors settle for the optimal t-sne hyper parameters for the current study? Could the process be generalized?

---

## Round 0.2 · Minor Revisions

The substantial improvements to the manuscript have rendered the manuscript acceptable to the reviewers. I read the manuscript once again and still found a few issues that need attention.

1. The key result is table 6 in which VGG is compared to the new ensemble models. The most important performance measure in diagnostic studies is AUC. The main p-value is 0.06 which is not significant. Therefore you should remove any statement that says this method outperforms or is better than the state of the art.
2. Careful inspection of table 6 reveals other issues. Comparing sensitivity and specificity is nonsense. Changing the decision threshold for exactly the same method (say VGG) will always create significant differences, but these differences obviously do not provide evidence of the method being better. Remove sensitivity and specificity as performance measures. Fail to see the relevance for F-score, MCC, MSE as performance measures. The best thing to do is to provide three ROC curves to provide visual evidence and then provide p-values. This is very common.
3. In the same table 6, the M1,M2,M3 comparisons are also randomly presented in arbitrary order. This makes this data in this table questionable. Provide a systematic presentation.
4. You removed the tSNE figures, but still a whole paragrahp in the results section describes tSNE details. The whole paragraph is incomprehensible and should be removed.
5. The results section contains a statistical analysis subparagraph which is a weird style and not at all compatible with ICMJE guidelines that you claimed to have read.
6. The abstract is not properly formatted in a sense that it provides an introduction, methods, results and conclusion section.

Reviewer 1 ·

Basic reporting

no comment

Experimental design

no comment

Validity of the findings

no comment

Additional comments

The authors addressed all the comments. The paper is easy to read, very interesting and presenting a model with great results. Congratulations!

·

Basic reporting

Good paper.

Experimental design

Accept

Validity of the findings

Accept

Additional comments

The paper has been revised well.

---

## Round 0.3 · Minor Revisions

A few minor edits, please.

I still see sensitivity and specificity as performance measures in the manuscript and the abstract while you agreed to remove it in your rebuttal. Remove these. BTW still weird to see a statistical analysis sub-section in the results section. Sloppy formatting.

In the abstract change the sentence: "It is observed that the ensemble model constructed with VGG-19 and SqueezeNet outperformed the state-of-the-art in all performance metrics toward classifying the parasitized and uninfected cells to aid in improved disease screening. " to ''It is observed that the ensemble model constructed with VGG-19 and SqueezeNet outperformed the state-of-the-art in several performance metrics toward classifying the parasitized and uninfected cells to aid in improved disease screening.''

Change the sentence: ''A similar trend is observed for AUC where the Ensemble-D model (0.9993±0.0004) outperformed VGG-19 (0.993±0.0006, p > 0.05)'' to ''A similar trend is observed for AUC where the AUC for Ensemble-D model and VGG-19 is 0.9993±0.0004) and 0.993±0.0006, p > 0.05) respectively'

---

## Round 0.4 · accepted · Accept

I agree with the final revisions.

#